# Berry By-Products in Combination with Antimicrobial Lactic Acid Bacteria Strains for the Sustainable Formulation of Chewing Candies

**DOI:** 10.3390/foods11091177

**Published:** 2022-04-19

**Authors:** Paulina Zavistanaviciute, Egle Zokaityte, Vytaute Starkute, Modestas Ruzauskas, Pranas Viskelis, Elena Bartkiene

**Affiliations:** 1Department of Food Safety and Quality, Veterinary Academy, Lithuanian University of Health Sciences, Tilzes Str. 18, LT-47181 Kaunas, Lithuania; egle.zokaityte@lsmuni.lt (E.Z.); vytaute.starkute@lsmuni.lt (V.S.); elena.bartkiene@lsmuni.lt (E.B.); 2Institute of Animal Rearing Technologies, Faculty of Animal Sciences, Lithuanian University of Health Sciences, Tilzes Str. 18, LT-47181 Kaunas, Lithuania; 3Department of Anatomy and Physiology, Faculty of Veterinary, Lithuanian University of Health Sciences, Tilzes Str. 18, LT-47181 Kaunas, Lithuania; modestas.ruzauskas@lsmuni.lt; 4Institute of Microbiology and Virology, Faculty of Veterinary, Lithuanian University of Health Sciences, Mickeviciaus Str. 9, LT-44307 Kaunas, Lithuania; 5Institute of Horticulture, Lithuanian Research Centre for Agriculture and Forestry, Kauno Str. 30, LT-54333 Babtai, Lithuania; biochem@lsdi.lt

**Keywords:** chewing candies, berry by-products, lactic acid bacteria, milk permeate, overall acceptability, emotions induced for consumers, antioxidant activity

## Abstract

The purpose of this research was to develop formulations of chewing candies (CCs) in a sustainable manner by using berry by-products in combination with antimicrobial lactic acid bacteria (LAB) strains. To implement this aim, the optimal quantities of by-products from lyophilised raspberry (Rasp) and blackcurrant (Bcur) from the juice production industry were selected. Prior to use, *Lactiplantibacillus plantarum* LUHS135, *Liquorilactobacillus*
*uvarum* LUHS245, *Lacticaseibacillus*
*paracasei* LUHS244, and *Pediococcus acidilactici* LUHS29 strains were multiplied in a dairy industry by-product—milk permeate (MP). The antimicrobial activity of the selected ingredients (berry by-products and LAB) was evaluated. Two texture-forming agents were tested for the CC formulations: gelatin (Gl) and agar (Ag). In addition, sugar was replaced with xylitol. The most appropriate formulation of the developed CCs according to the product’s texture, colour, total phenolic compound (TPC) content, antioxidant activity, viable LAB count during storage, overall acceptability (OA), and emotions (EMs) induced in consumers was selected. It was established that the tested LAB inhibited three pathogens out of the 11 tested, while the blackcurrant by-products inhibited all 11 tested pathogens. The highest OA was shown for the CC prepared with gelatin in addition to 5 g of Rasp and 5 g of Bcur by-products. The Rasp and LUHS135 formulation showed the highest TPC content (147.16 mg 100 g^−1^ d.m.), antioxidant activity (88.2%), and LAB count after 24 days of storage (6.79 log_10_ CFU g^−1^). Finally, it was concluded that Gl, Rasp and Bcur by-products, and *L. plantarum* LUHS135 multiplied in MP are promising ingredients for preparing CCs in a sustainable manner; the best CC formula consisted of Gl, Rasp by-products, and LUHS135 and showed the highest OA (score 9.52) and induced the highest intensity of the EM ‘happy’ (0.231).

## 1. Introduction

Gummy or jelly candies are foods that are in quite high demand; they are consumed by a large and miscellaneous group of people from children to elders, despite the fact that one of their main ingredients is sugar, which is usually incorporated in the form of sucrose syrup and/or glucose [1,2,3]. The high content of sugar and food additives incorporated in the formulas of sweets has a negative impact on human health; despite this, their consumption is increasing [2]. The high consumption of sugar has a negative impact on children’s health, for instance, increasing obesity, impulsiveness, addictive behaviour, and stress-driven anxiety [4]. To improve the healthy aspects of gummy or jelly candies, natural sweeteners can be added to replace sugar [5]. One of these alternatives is xylitol—a low-calorie, crystalline sweetener which is naturally present in fibrous plant foods and hardwood trees [6,7]. Xylitol has several potential health benefits, including being low in calories; having insulin-independent metabolism, a prebiotic nature, and anabolic effects; and being safe to use [6,7]. Gummy or jelly candies can be prepared with xylitol, since it imparts a quick sweetness, flavour, and a cooling effect [8].

The use of fruit and berry products, as well as their by-products, has been considered for the preparation of gummy and jelly candies in a sustainable manner [1,9,10]. These ingredients can improve the sensory properties (colour, flavour, chewiness, adhesiveness, and texture) of gummy and jelly candies; furthermore, their addition generates healthier formulations with compounds possessing desirable antioxidant and antimicrobial properties [1,2,5,7,10]. For instance, blackcurrant (*Ribes nigrum* L.; Bcur) is a natural source of anthocyanins, the amount of which varies from 1260 to 2878 mg 100 g^−1^ (dry weight) depending on the genotype. Bcur anthocyanins can replace synthetic colourants and enrich food products with bioactive ingredients [11,12]. In addition to anthocyanins, Bcur contains phenolic acid derivatives and hydroxybenzoic and hydroxycinnamic acids [13], which possess antimicrobial and antifungal activity [13]. Bcur flavonols (glycosides of myricetin, quercetin, kaempferol, and isorhamnetin [13]) show potent activity against free radicals even at low concentrations [14,15]; they also possess anti-proliferative activity against human acute leukaemia HL-60 cells [16]. Raspberry fruits (*Rubus idaeus* L.; Rasp) also have a high content of valuable bioactive compounds, including anthocyanins and flavonols, phenolic acids (ellagic, gallic, chlorogenic, p-coumaric, caffeic, hydroxybenzoic, and protocatechuic acids [17]), and vitamin C [12]. A profuse number of epidemiological studies have shown that there is a correlation between raspberry consumption and the evolution of various diseases: raspberries reduce the risk of cardiovascular disease, diabetes, and different types of cancer [13,18,19,20,21,22]. Moreover, the use of large-dose antioxidants, such as vitamin C, can be an effective treatment for COVID-19 (SARS-CoV-2) patients [13,23]. However, a very high quantity of these beneficial compounds are lost since they remain in the Bcur and Rasp by-products [7,24]. From this point of view, gummy or jelly candy formulations based on berry by-products could be a valuable source of bioactive compounds.

To improve the quality of confections, especially gummy or jelly candies, lactic acid bacteria (LAB) can be used [25]. However, LAB are susceptible to their environmental surroundings, as well as thermal processing, which can have a negative impact on their application on an industrial scale [26,27]. Even so, the viability of LAB during storage can be improved by lyophilisation [7,27]. To ensure an environmentally friendly approach, LAB can be multiplied and later freeze-dried in food industry by-products, including milk permeate (MP) [28], cheese whey [26], apple pomace [29], etc. In this study, MP was selected as a potential substrate for multiplying LAB. The MP obtained during the production of milk protein concentrate is sterile and contains lactose; the latter, during fermentation, can be converted to galacto-oligosaccharides (GOS) [28,30]. GOS are desirable compounds in food due to their prebiotic properties, and have a certain impact on several health parameters; for instance, they stimulate the growth of *Bifidobacteria* and *Lactobacillus* species in the intestinal tract, which inhibit and bind *E. coli*, *Salmonella typhimurium*, and *Clostridia* [3,31]. They also induce the immune system indirectly via the production of antimicrobial materials, which reduce the proliferation of pathogenic bacteria [31,32], and have an antitumor effect [33]. Moreover, MP fermentation with LAB strains can generate a higher quantity of GOS, due to lactose conversion by LAB [28,30]. LAB that have been multiplied in a safe substrate (i.e., MP) and lyophilised can be used to provide desirable antimicrobial properties for nutraceutical chewable candies, to ensure a positive effect on consumers’ health [7,9,10,34]. It is known that LAB have health-promoting effects, such as protection against infectious agents, immunomodulatory effects, anti-allergenic effects, anti-obesity effects, antioxidant effects, the ability to enhance the bioavailability of vitamins or minerals, etc. [3,35,36,37,38]. In this study, we hypothesised that selected LAB (multiplied in MP and lyophilised) and by-products from the berry juice production industry (lyophilised Rasp and Bcur), as well as xylitol, could be used for the sustainable preparation of added-value chewing candies.

The aim of this study was to develop formulations of chewing candies (CCs) in a sustainable manner using by-products from the berry juice production industry (Rasp and Bcur) in combination with antimicrobial LAB strains, technologically functionalised in MP. In addition, in the CC formulations, sugar was replaced with xylitol.

## 2. Materials and Methods

### 2.1. Materials Used for the Preparation of CCs

Agar powder (*Gelidium sesquipedale* algae; Ag) and gelatin (Gl) were purchased from Rotmanka (Gdansk, Poland) and Klingai (Lithuania), respectively. Ag powder forms a high-strength gel at low concentrations, is transparent, and has a low viscosity in solution [39]. Gl forms a gel with a cohesive internal structure, and has a higher viscosity, melting temperature, and setting temperature [40].

Lyophilised Rasp (*Rubus idaeus*, variety ‘Poliana’) and Bcur (*Ribes nigrum*, variety ‘Ben Alder’) by-products were obtained from the Institute of Horticulture, Lithuanian Research Centre for Agriculture and Forestry (Babtai, Kaunas distr., Lithuania) in 2021. The Rasp and Bcur by-products, consisting of the seeds, peels, and fibres, were generated during juice production.

Xylitol (Natur Hurtig, Nuremberg, Germany) and citric acid (Sanitex, Kaunas, Lithuania) were purchased from a local market (UAB ‘Maxima LT’, Kaunas, Lithuania).

*Lactobacillus plantarum* LUHS135, *L. uvarum* LUHS245, *L. paracasei* LUHS244, and *Pediococcus acidilactici* LUHS29 strains, previously isolated from spontaneously fermented cereals, were selected for CC preparation according to their desirable antimicrobial and antifungal properties [41].

MP was obtained from the ‘Pienas LT’ agricultural cooperative (Biruliskes, Lithuania) and used for LAB multiplication. Lyophilised biomass was used for CC preparation. During fermentation, LAB convert lactose to prebiotic GOS; however, information about the physicochemical parameters of MP is scarce [28].

### 2.2. LAB Multiplication in MP and Stabilisation

LUHS135, LUHS244, LUHS245, and LUHS29 strains were isolated from spontaneously fermented cereal, as described by Bartkiene et al. [41]. Before the experiment, LAB strains were stored at −80 °C (PRO-LAB Diagnostics, Bromborough, UK), complemented with 20% glycerol. Prior to the experiment, the selected LAB strains were multiplied in MRS broth (CM 0359, Oxoid Ltd., Hampshire, UK) at 30 °C for 48 h.

MP was stored at −18 °C before use. A total of 3% LAB (with a cell concentration of 9.2 log_10_ CFU mL^−1^) was inoculated in MP (*v*/*v*), followed by anaerobic fermentation in a modified atmosphere of CO_2_ in a chamber incubator (Memmert GmbH + Co. KG, Schwabach, Germany) for 48 h at 30 °C. The method was described in detail by Zokaityte et al. [28].

LAB strains multiplied in MP were lyophilised. For LAB biomass lyophilisation, a 3 × 4 × 5 sublimator (ZIRBUS Technology GmbH, Bad Grund, Germany) was used. The condenser temperature was −85 °C, the vacuum was 2 × 10^−6^ mPa, and the samples were frozen at −40 °C in a laboratory freezer then left in the freeze dryer for 72 h [27]. Lyophilised LAB biomass was used for CC preparation.

### 2.3. Principal Scheme of the Experiment and Formulations of CC Recipes

The principal scheme of the experimental design is shown in Figure 1a,b. The first stage of the experiment (Figure 1a) evaluated the antimicrobial activity of the separate ingredients: fermented MP (MP_LUHS135_, MP_LUHS245_, MP_LUHS244_, and MP_LUHS29_) and Rasp and Bcur by-products. Different gel-forming agents and different amounts of Rasp and Bcur by-products were also tested (Table 1).

For the preparation of CCs with Ag, firstly, Ag powder was soaked in water for 30 min at a temperature of 30 ± 2 °C and melted by heating for 5 min to a temperature of 100 °C. Thereafter, the other ingredients (Table 1) were added and mixed under boiling conditions (100 °C) with stirring [9]. For the preparation of CCs with Gl, firstly, Gl powder was soaked in water for 30 min at a temperature of 30 ± 2 °C and melted at 80 ± 2 °C; thereafter, the other ingredients (Table 1) were added and mixed in. After mixing, the mass with Ag and Gl was poured into a mould, and the CCs were dried at 22–24 °C for 24 h to obtain a hard gel form [9]. In the second stage of the experiment (Figure 1b), in addition to the chosen ingredients, lyophilised LAB powder was added (Table 2). The preparation of the improved CCs was performed in the same way as described above; in addition, lyophilised LAB powder was added at 38 ± 2 °C and the mass was mixed.

For the prepared CCs, the texture, colour coordinates, and overall acceptability (OA) were tested. According to the most appropriate results, Gl as the gel-forming agent and 5 g of Rasp and Bcur by-products were selected for further CC formulations.

For the prepared CC samples, the texture, colour coordinates, total phenolic compound (TPC) content, antioxidant activity, viable LAB cell count during storage, OA, and emotions (EMs) induced in consumers were evaluated.

### 2.4. Evaluation of the Antimicrobial Activity of LAB Multiplied in MP and Berry By-Products

The antimicrobial activity of LAB multiplied in MP and Rasp and Bcur by-products was tested via an agar well diffusion assay [41]. For this purpose, a 0.5 McFarland unit density suspension (~10^8^ CFU mL^−1^) of each pathogenic bacterial strain was inoculated onto the surface of cooled Mueller–Hinton agar (Oxoid, Basingstoke, UK) using sterile cotton swabs [42]. Wells of 6 mm in diameter were punched in the agar and filled with 50 μL of LAB multiplied in MP and 50 μg of Rasp and Bcur by-products [41,42]. The antimicrobial activity against the tested bacteria was determined by measuring the diameter of the inhibition zones (mm).

### 2.5. Analysis of CC Colour Characteristics and Texture

The colour coordinates (L*, a*, b*) were measured using a CIELAB system (Chromameter CR-400, Konica Minolta, Tokyo, Japan).

Texture (hardness) was analysed using a Brookfield CT-3 Texture Analyzer (Middleboro, MA, USA).

### 2.6. Determination of TPC Content and the Antioxidant Activity of Prepared CCs

The TPC content of the CC samples was determined by the spectrophotometry method described by Vaher et al. [43]. A total of 0.2 mL of every fraction of free phenolics was blended with 1 mL of Folin–Ciocalteau reagent and 0.8 mL of a saturated Na_2_CO_3_ solution. The prepared mixed solution was stored at room temperature (25 °C) for 30 min and then the absorbance was measured at 765 nm with a J.P. SELECTA S.A. V-1100D spectrophotometer (Barcelona, Spain); TPC content was expressed as microgram of gallic acid equivalent mL^−1^ of solution [43].

The ability of the CC extract to scavenge DPPH free radicals was assessed using the standard method described by Zhu et al. [44]. According to the protocol, 2 mL of DPPH solution (0.1 mM in ethanol) was blended with 2 mL of the samples dissolved in the extracting solvent. The solution was shaken and incubated in the dark at room temperature (25 °C) for 60 min, and then the absorbance was measured at 517 nm with a J.P. SELECTA S.A. V-1100D spectrophotometer (Barcelona, Spain). The inhibition of the DPPH radicals by the sample was calculated according to the following formula [44]:$$\mathrm{DPPH}\;\mathrm{scavenging}\;\mathrm{activity}\;\left(\%\right)=\frac{\mathrm{Absorption}\;\mathrm{of}\;\mathrm{control}-\mathrm{absorption}\;\mathrm{of}\;\mathrm{sample}}{\mathrm{Absorption}\;\mathrm{of}\;\mathrm{control}}\times 100\%\;$$

### 2.7. Determination of Viable LAB Count in CC Formulations during Storage

To evaluate the viable LAB count in the CC samples, 10 g of each sample was homogenised with 90 mL of saline (0.9%). Viable LAB counts were determined on MRS agar (Liofilchem, Roseto degli Abruzzi, Teramo, Italy) using standard plate count techniques. The plates were incubated at a temperature of 30 °C for 72 h under anaerobic conditions (using an AnaeroGen atmosphere generation system, Oxoid, Basingstoke, UK). The viability of LAB in the CC samples was determined after 24 h, 14 days, and 24 days of storage at 18 ± 2 °C. All results were expressed in log_10_ CFU g^−1^, as described by Zokaityte et al. [28].

### 2.8. Evaluation of OA and EMs Induced in Consumers by CC

The OA of the CCs using Gl as the gel-forming agent and containing lyophilised LAB was assessed by 30 judges, according to ISO method 8586-1 [45], using a 10-point scale ranging from 0 (‘dislike extremely’) to 10 (‘like extremely’) [45].

The prepared CCs were tested by 30 judges by applying the FaceReader 8.0 software (Noldus Information Technology, Wageningen, The Netherlands), scaling eight different emotional patterns (neutral, happy, sad, angry, surprised, scared, disgusted, and contempt) and valence (scores ranged from −1 to 1). The procedure was described in detail by Bartkiene et al. [46].

### 2.9. Statistical Analysis

The results were expressed as the mean ± standard deviation (SD). The preparation of the CCs was performed once, and all analyses were performed in triplicate. The results were analysed using the SPSS statistical package for Windows V15.0 (SPSS Inc., Chicago, IL, USA, 2007). In order to evaluate the influence of different factors (different gel-forming agents, the quantity of berry by-products, different LAB strains, and their interaction) on the analysed parameters of the CCs, a multivariate analysis of variance (ANOVA) was performed and a Tukey HSD test was used as a post-hoc test (statistical program R3.2.1, R Core Team, 2015). In addition, a linear Pearson’s correlation was used to quantify the strength of the relationship between the variables.

The significance of differences between the samples was evaluated using Tukey range tests at a 5% level. The results were considered as statistically significant at *p* ≤ 0.05.

## 3. Results

### 3.1. Evaluation of the Antimicrobial Activity of LAB Multiplied in MP and Berry By-Products

The antimicrobial activity of the tested LAB multiplied in MP and berry by-products is presented in Table 3. The growth of *Streptococcus mutans* and *Pasteurella multocida* was inhibited by all the tested LAB strains. The largest diameter of inhibition zone (DIZ) against *S. mutans* was observed for the LUHS244 and LUHS245 samples (15.4 ± 0.2 and 15.4 ± 0.3 mm, respectively). The largest DIZs against *P. multocida* were established for LUHS135, LUHS244, and LUHS245 (20.4 ± 0.6, 20.2 ± 0.9, and 20.7 ± 0.6 mm, respectively). The growth of *Staphylococcus epidermis* was suppressed by LUHS245 and LUHS244, with the largest DIZ observed for LUHS245 (13.0 ± 0.6 mm). Only LUHS135 inhibited the growth of *S. haemolyticus* (DIZ 10.2 ± 0.6 mm). The antimicrobial activity of LUHS135, LUHS245, LUHS244, and LUHS29 is shown in Appendix A; our previous studies showed that LUHS135, LUHS244, LUHS245, and LUHS29 counts in MP were 8.58 ± 0.24, 8.06 ± 0.31, 8.68 ± 0.39, and 8.19 ± 0.23 log_10_ CFU g^−1^, respectively, after multiplication for 48 h at 30 °C [28].

*L. uvarum* is the part of the *Lactobacillus salivarius* group [47]. Bartkiene et al. [42] recently reported that *L. uvarum* has high potential as an antimicrobial ingredient for the preparation of food coatings due to its ability to inhibit a broad spectrum of Gram-positive and Gram-negative pathogenic and opportunistic strains. *L. plantarum* is known for its ability to produce several natural antimicrobial substances, such as bacteriocins; propionic, lactic, and acetic acids; and hydrogen peroxide [27]. *L. plantarum* produces the bacteriocin plantaricin, which shows high antimicrobial activity against Gram-negative bacteria, such as *E. coli*, and Gram-positive bacteria, such as *Staphylococcus aureus* [48]. *P. acidilactici* produces pediocin and pediocin-like bacteriocins, which demonstrate a broad spectrum of antimicrobial activity against Gram-positive bacteria, especially *Listeria monocytogenes* [49,50]. The antimicrobial effect can be explained by the formation of pores in the cytoplasmic membrane and cell membrane dysfunction [50,51]. According to Iseppi et al. [52], *L. paracasei* and *L. brevis* produce class II bacteriocins, described as a group of small, heat-stable non-lantibiotics and low-molecular-weight proteins that are already recognised as being active against some human pathogens, including *S. aureus*, *S. sanguis*, and *Pseudomonas aeruginosa* [53,54]. Our results show that LAB biomass multiplied in MP is a promising food ingredient for its antimicrobial properties.

According to our results, Bcur by-products inhibited the growth of all the tested pathogenic and opportunistic bacteria (Table 3). Similar tendencies were found in Rasp by-products; however, they did not inhibit the growth of *Salmonella enterica*. The largest DIZs were found for Bcur and Rasp by-products against *P. multocida* (30.7 ± 0.5 and 30.6 ± 0.4 mm, respectively). The largest DIZs against *P. aeruginosa*, *Enterococcus faecalis*, *E. faecium*, *Bacillus cereus*, and *S. haemolyticus* (16.2 ± 0.1, 12.5 ± 0.1, 14.3 ± 0.3, 21.4 ± 0.2, and 18.1 ± 0.4 mm, respectively) were found for Bcur by-products. Opposite tendencies were established for Rasp by-products, with the largest DIZ being observed against MRSA and *S. mutans* (15.1 ± 0.3 and 25.0 ± 0.5 mm, respectively).

Our study results are in agreement with those of Bartkiene et al. [42] showing that lyophilised berry by-products possess high antimicrobial activity. Bcur (*R. nigrum* L.) by-products have a high amount of polyphenols, particularly anthocyanins, phenolic acid derivatives, and flavanols, as well as proanthocyanins, which show desirable antioxidant and antimicrobial properties [55]. The antimicrobial outcome of polyphenolic compounds has been described in several studies [56,57,58]. In addition to this, Bcur juice and extracts have an intense effect on blood glucose levels, delaying blood glucose responses and reducing peak glucose during oral glucose tolerance tests by inhibiting carbohydrate absorption during the early phase (0–30 min) [59]. According to Cook et al. [60], 600 mg day^−1^ of anthocyanins from a Bcur extract (consumption period of 7 days) resulted in a reduction in systolic (6 mmHg) and diastolic (6 mmHg) blood pressure in healthy older adults [59]. These results led to the conclusion that these changes improve health by reducing stroke and coronary heart disease mortality by 35–40% and 20–25%, respectively [61]. Raspberries (*R. idaeus*) have a high amount of anthocyanins, as well as ellagitannins, conjugates of ellagic acid, and quercetin, which show antimicrobial activity against human intestinal tract pathogens [62]. Raspberries inhibit the growth of *Helicobacter pylori* and *B. cereus*, while the inhibition of *Campylobacter jejuni* and *Candida albicans* is connected to the high amount of ellagitannins in the extracts [63,64]. According to Bauza-Kaszewska et al. [65], the combination of *Lactobacilli* and Rasp is synergic against *Salmonella enteritidis*, *S. typhimurium*, and *L. monocytogenes*. Bartkiene et al. [42] found that the combination of Rasp by-products with *L. casei* or *L. uvarum* resulted in increased antimicrobial activity against Gram-positive (*S. aureus*, *E. faecalis*, *E. faecium*, *B. cereus*, and *S. mutans*) and Gram-negative bacteria (*Klebsiella pneumoniae*, *S. enterica*, *P. aeruginosa*, and *Proteus marabilis*). According to Kirakosyan et al. [66], the dietary intake of Rasp can decrease cardiometabolic risk and reduce left ventricular enlargement and the thickening of the heart wall.

### 3.2. Evaluation of the Texture, Colour Coordinates, and OA of CC Prepared with Different Amounts of Berry By-Products and Different Gel-Forming Agents

The texture (hardness), colour coordinates (L*—lightness; a*—redness; b*—yellowness; NBS), and OA of CCs made with Ag and Gl and different quantities of berry by-products are presented in Table 4. The results of the multivariate ANOVA showed that there was a significant (*p* ≤ 0.05) effect of selecting Ag and/or Gl and a different kind of berry by-product on hardness, colour coordinates, and OA. According to our results, all CCs prepared with Gl were softer (by 86%, in comparison to the samples prepared with Ag). The softest texture was found for the CC5.0raspGl samples (0.700 ± 0.01 mJ), while the hardest samples were CC5.0bcurAg (10.1 ± 0.5 mJ). Our findings are in contrast with the results published by Zokaityte et al. [9], who found that gummy candies prepared with Ag with added psyllium husk and apple pomace were 40% softer than samples prepared with Gl. This can be explained by psyllium husk having a sucrose content of less than 0.1 g 100 g^−1^; the quantities of sugar can influence the structure formation of gummies with Ag [67]. According to Elawady et al. [68], the viscosity, elasticity, and hardness of Ag gels depends on the sugar concentration and the characteristics of plant-based ingredients that are incorporated in the gummy formulations. The formation of Gl gels depends on the quantities of Gl, sugars, and polyols added, and the amount of pectin in apples could also be a relevant factor [69]. In our study, the Rasp and Bcur by-products used had a low concentration of pectin, and this characteristic could lead to the softer texture of CCs prepared with Gl. In contrast, the natural sugars in the berry by-products could lead to a harder texture of CCs prepared with Ag. The hardness of gummy candies is related to chewiness and gumminess. In addition to this, a harder gummy candy texture leads to a firmer candy structure [70]. Texture has an essential influence on the acceptability of foods, as the textural attributes perceived by the mouth influence the joy of eating [70,71].

A strong negative correlation (r = −0.8663) was found between the texture and OA of the CCs. The highest OA (score 9.43 ± 0.3 and 9.20 ± 0.4) was shown for samples whose texture was formed with Gl (CC5.0raspGl and CC5.0bcurGl, respectively). It should be mentioned that CCs prepared with Gl, in all cases, had a higher OA than those prepared with Ag. Similar tendencies were reported by Zokaityte et al. [9] and Lele et al. [10]. OA and sensory evaluation are two of the most important properties of food; however, to be successful, these properties must be in relation to the physical, chemical, formulation, and process variables [70,72].

The colour of food can be one of the main factors in the understanding of quality by consumers, as it greatly affects sensory acceptance [73]. In our study, the lightness (L* coordinate) of prepared CCs varied from 27.9 ± 0.26 to 42.4 ± 0.14 NBS. Differing the quantity of the berry by-products did not have a significant effect on the lightness of the prepared CCs; however, this factor had a significant effect (*p* < 0.001) on the redness (a*) and yellowness (b*) colour coordinates of the tested samples. The lowest a* value was found in all samples prepared with Bcur by-products and varied from 1.36 ± 0.02 to 7.95 ± 0.12 NBS. The a* colour coordinate of samples prepared with Rasp by-products was more than 5.2 times higher, and results varied from 19.83 ± 0.17 to 25.8 ± 0.57 NBS. The same tendency was found for the b* colour coordinate of the CCs, the lowest being found for the CC made with Bcur by-products, where the results varied from 1.18 ± 0.02 to 1.92 ± 0.03 NBS. The b* colour coordinate of the CCs prepared with Rasp by-products was more than 7.8 times higher. These findings can be interpreted by the different colour of the Rasp and Bcur by-products.

Our results from this stage of the experiment showed that CCs made with 5.0 g of Bcur and Rasp by-products and Gl had better texture properties and a higher OA. Thus, in the next stage, these CC compositions were enriched with lyophilised LAB (which were multiplied in MP) to improve the quality parameters of the final product.

### 3.3. Evaluation of the Texture and Colour Coordinates of CCs Containing LAB

The texture parameters (mJ) and colour coordinates (L*—lightness, a*—redness, and b*—yellowness; NBS) of CCs made with different LAB strains are presented in Table 5. The results of the multivariate ANOVA indicated that the effect of the berry by-product and LAB strain used on the texture parameters of CCs made with LAB was significant (*p* < 0.001). The samples with the hardest texture were Rasp29 and Bcur29 (1.91 ± 0.01 and 2.10 ± 0.01 mJ, respectively), and the softest were Rasp135 and Bcur135 (1.10 ± 0.02 and 0.8 ± 0.01 mJ, respectively). Comparing this with the texture results described in Section 3.2, we can state that the addition of LUHS245, LUHS244, and LUHS29 increased the hardness of the CCs, while LUHS135 resulted in samples with a softer texture. The texture of gummy or nutraceutical candies is one of the most important parameters. According to Altınok et al. [74], the particle size of the different additives (plant by-products, prebiotics, or probiotics) has a significant influence on the texture parameters, including hardness, adhesiveness, chewiness, and resilience, of gummy candies. A weak negative correlation (r = −0.4039) was found between the texture and OA of CCs containing LAB.

The results of the multivariate ANOVA indicated that the effect of the berry by-product and LAB strain used on the colour coordinates (L*, a*, b*) of CCs made with LAB was significant (*p* < 0.0001). The L* colour coordinate (lightness) of CCs containing LAB varied from 27.0 ± 0.06 to 37.6 ± 0.15 NBS (Table 5). Similar tendencies were observed for the a* and b* colour coordinates of CCs containing LAB. The lowest a* and b* colour coordinates were found in all samples prepared with Bcur by-products, varying from 2.07 ± 0.14 to 2.91 ± 0.09 NBS and from 1.34 ± 0.02 to 1.54 ± 0.03 NBS, respectively. The highest a* and b* colour coordinates were found in all samples prepared with Rasp by-products, varying from 17.8 ± 0.18 to 21.7 ± 1.33 NBS and from 10.0 ± 0.10 to 10.7 ± 0.21 NBS, respectively. When comparing the colour coordinates according to the LAB used, the lowest a* coordinate was determined for the Bcur244 sample (2.07 ± 0.14 NBS), while the highest was found for Rasp135 (21.7 ± 1.33 NBS). The lowest b* coordinates were found for Bcur29, Bcur135, and Bcur244 samples (1.35 ± 0.03, 1.36 ± 0.03, and 1.34 ± 0.02 NBS, respectively); the highest b* colour coordinate was established for Rasp244 (10.74 ± 0.21 NBS). According to Figueroa and Genovese [67], prebiotic- or probiotic-enriched candies are darker and the colour itself depends on the colour of the additives.

### 3.4. TPC Content and Antioxidant Activity of CCs Containing LAB

The TPC content (mg 100 g^−1^ d.m.) and antioxidant activity (%) of the prepared CCs are shown in Table 6. The highest TPC content was found in the Rasp135 sample (147.1 ± 0.13 mg 100 g^−1^ d.m.). The other CC samples showed, on average, a 21.8% lower TPC content, with the lowest found for Bcur245 (84.1 ± 0.15 mg 100 g^−1^ d.m.). Similar trends were observed for the antioxidant activity of the CCs, the highest being found in three CC samples (Rasp135, Rasp245, and Bcur29: 88.2 ± 1.34%, 86.1 ± 0.95%, and 87.0 ± 1.02%, respectively; Table 6). The other CC samples displayed, on average, a 11.9% lower antioxidant activity, with the lowest found for Bcur244 (65.3 ± 0.77%). In addition to this, the results of the multivariate ANOVA indicated that there was a significant (*p* < 0.0001) impact of the berry by-product and LAB strain used, as well as the interaction of both factors, on the TPC content and antioxidant activity of CCs containing LAB. In addition, a moderate positive correlation (r = 0.6462) was found between TPC content and antioxidant activity.

The improvement of sweet recipes, based merely on natural components with antioxidant properties, can provide advantageous solutions to the confectionery industry, thus reducing the amount of synthetic additives [1]. Our findings are in agreement with those of other researchers showing that antioxidant activity is positively correlated with TPC content [1,75]. The strong relationship between TPC content and free radical scavenging activity can be explained by the combined impact of the different phenolic compounds in Rasp and Bcur by-products and their strong hydrogen atom-donating ability [1,75]. In addition, the use of citric acid contributes to a higher antioxidant activity and TPC content, while also improving the colour of jelly or gummy candies [1]. Moreover, the antioxidant activity of red Rasp indicates that they can eliminate free radicals, such as DPPH and ABTS [76], due to their ferric- and cupric-reducing antioxidant potential, in addition to their ability to prohibit β-carotene discoloration [77,78,79]. According to Lele et al. [10], LAB have a significant effect on the antioxidant activity of gummy candies prepared with Gl and prebiotics. For instance, substrate fermentation with *L. plantarum* can increase the antioxidant activity and TPC count by 10% compared with samples without fermentation [80]. These results can be explained by the ability of *L. plantarum* to break down the ester bonds in plants, liberating free phenolic compounds that effect antioxidants via several separate mechanisms, including free radical scavenging capacity [81], the chelation of metal ions [82], and the inhibitory activity of prooxidant enzymes [80,82]. Antioxidants present in berries may have a significant effect on the prophylaxis and progression of various diseases associated with oxidative stress pathologies, such as cardiovascular diseases, inflammation, cancer [83], and neurodegenerative diseases [84]. Berries and their products (juice, extracts, etc.) have a significant impact as antioxidants in humans in both in vitro and in vivo models using dietary complementation with different berries [84,85]. Cano-Lamadrid et al. [1] found that using Gl as the gel-forming agent reduced the antioxidant activity of jelly candies, and the findings of Zokaityte et al. [9] are in agreement with these results. However, the antioxidant activity and TPC content of CCs prepared with Gl can be improved by including Rasp and Bcur by-products, citric acid, and LAB in the formulation, as these ingredients have the above-mentioned properties.

### 3.5. LAB Count in CCs during Storage

The LAB count in CC samples after 24 h, 14 days, and 24 days of storage is presented in Figure 2. Comparing the LAB counts in the CCs after 24 h, the highest were found in the Rasp135 and Rasp29 samples (8.12 ± 0.06 and 8.05 ± 0.07 log_10_ CFU g^−1^, respectively). A LAB count higher than 7.0 log_10_ CFU g^−1^ was established after 24 h in five CC samples (Rasp244, Rasp245, Bcur244, Bcur135, and Bcur29). The lowest LAB count after 24 h, 6.99 ± 0.06 log_10_ CFU g^−1^, was observed in Bcur245. Similar tendencies of LAB counts in CCs were found after 14 and 24 days of storage. The highest LAB count after 14 days of storage, 7.53 ± 0.05 log_10_ CFU g^−1^, was found in Rasp135; however, the results for this sample decreased by 7.2% compared with those after storage for 24 h. The lowest LAB count after 14 days of storage, 6.51 ± 0.06 log_10_ CFU g^−1^, was observed in Bcur29, and the results for this sample decreased by 9.5% compared with those after storage for 24 h. After 14 days of storage, the LAB counts in all CC samples were reduced, by 5.8% on average. The highest LAB counts after 24 days of storage, 6.85 ± 0.10 and 6.79 ± 0.10 log_10_ CFU g^−1^, were found in Bcur135 and Rasp135, respectively, and the results for these samples decreased by 8.0% and 16.4%, respectively, compared with those after storage for 24 h. The lowest LAB count after 24 days of storage, 5.91 ± 0.03 log_10_ CFU g^−1^, was observed in Bcur245, and the results for this sample decreased by 14.3% compared with those after storage for 24 h. After 24 days of storage, the LAB count in all CC samples was reduced by 16.8% on average.

LAB are generally recognised as safe (GRAS) in the USA and several LAB strains fulfil the criteria of the qualified presumption of safety (QPS) in Europe [86]. Despite their desirable characteristics, such as antimicrobial activity and the improvement of sensory, quality, or safety parameters in food matrices, LAB are sensitive to environmental conditions [87,88]. One alternative to improve LAB stability during storage is immobilisation in polymeric substances, including Gl [89,90], Ag [91,92], or chitosan [93]. Immobilisation prolongs the activity and stability of the immobilised LAB cells, since the immobilisation material acts as a protective agent against physicochemical changes, such as those occurring with pH, temperature, and bile salts, and improves viability during storage [94]. According to Zokaityte et al. [9], Gl can be recommended for extending LAB viability in nutraceutical CCs (the term “nutraceutical” is used to describe medicinally or nutritionally functional foods) because, after 14 days of storage, they found LAB counts higher than 6.0 log_10_ CFU g^−1^ in samples prepared with Gl. Another alternative to improve LAB stability during storage is freeze drying. According to Bartkiene et al. [7], LAB can be encapsulated in acid whey media, and different additives, such as glucose, sucrose, or yeast extract, can increase bacterial cell viability and stability during storage and can be incorporated in the CC formula to improve the desirable characteristics of the final product. In our study, we applied both technologies (lyophilising LAB in MP and later using them in a CC formula with Gl) to improve the stability of LAB in the final product during storage. Our findings are very important because CCs possessing more than 6.0 log_10_ CFU g^−1^ viable LAB cells can have probiotic properties.

### 3.6. OA and EMs Induced in Consumers by CCs

The OA and EMs induced in consumers by the prepared CCs are shown in Table 7. The highest OA scores (9.52 ± 0.10 and 9.39 ± 0.20) were shown for the Rasp135 and Bcur135 samples, respectively. The lowest OA (score 5.30 ± 0.51) was shown for Bcur245. Comparing OA results between the berry by-products used, a 14.7% higher OA, on average, was obtained for CC samples prepared with Rasp by-products. In addition, it should be mentioned that CCs prepared with LUHS135 had the highest OA scores of all the tested CC samples.

Comparing the EMs induced in consumers by the prepared CCs, the highest score for the EM ‘happy’ was found when consumers were testing the Rasp135 sample (0.231 ± 0.003), in addition to it having the highest OA score, and a very strong positive correlation was found between OA and the EM ‘happy’ (r = 0.7708) and between OA and ‘valence’ (r = 0.7976). The lowest score for the EM ‘happy’ was found when consumers were testing Bcur245 (0.023 ± 0.001), which had the lowest OA score. Samples prepared with Rasp by-products elicited a 66.1% higher score for the EM ‘happy’, on average, compared with CC samples prepared with Bcur by-products. A moderate negative correlation was found between OA and the EM ‘sad’ (r = −0.523) and between OA and ‘angry’ (r = −0.5974). The lowest score for the EM ‘sad’ was found when consumers were testing the Rasp135 and Bcur29 samples (0.081 ± 0.010 and 0.090 ± 0.030, respectively). The highest score for the EM ‘sad’ was found when consumers were testing Rasp244 (0.236 ± 0.006). The lowest score for the EM ‘angry’ was found when consumers were testing Rasp135 and Rasp29 (0.055 ± 0.001 and 0.050 ± 0.004, respectively). The highest score for the EM ‘angry’ was found when consumers were testing Rasp245 (0.233 ± 0.003). The scores for the EMs ‘surprised’, ‘scared’, and ‘disgusted’ for all tested CC samples varied on average by 0.004, 0.003, and 0.003, respectively. The highest score for the EM ‘contempt’ was found for Rasp135 (0.100 ± 0.010), while the lowest was found for Bcur29 (0.001 ± 0.0002). The lowest intensity of the EM ‘neutral’ was expressed while consumers were testing Bcur245 (0.120 ± 0.002). The highest intensity of the EM ‘neutral’ was shown when Rasp29 and Bcur29 samples were tested (0.622 ± 0.031 and 0.651 ± 0.040, respectively). According to the OA and EM results, Rasp by-products and *L. plantarum* LUHS135 showed higher scores compared with Bcur by-products in combination with other LAB strains, except for *L. plantarum* LUHS135.

It is known that foods cause a wide variety of Ems, and these factors usually have a significant effect on consumer choice. Negative expressions are shown more accurately compared with positive EMs due to limited facial motion [9,88,95,96]. In this study, EM expression showed that the positive EMs (especially ‘happy’) were more intense than the negative EMs (e.g., ‘angry’, ‘sad’, or ‘disgusted’). These results are in agreement with other published studies reporting that consumers use more positive EMs to describe nutraceutical candies than negative ones [9,88]. It is known that smiling people express happiness; however, a smile can also show disappointment or hide real EMs [97,98]. However, in this study, there was a strong positive correlation between the expression of the EM ‘happy’ and OA results. The investigation of emotional reactions to food products could provide further information about consumers’ choice of food and the acceptability of new products [99]. According to Cano-Lamadrid et al. [1], if consumers are aware of the nature and ingredients in the jelly they are tasting, it can lead to higher scores of OA.

## 4. Conclusions

The developed CCs could be a healthier alternative to traditional gummy or jelly candies and their production could lead to the sustainable valorisation of dairy and juice industry by-products. The highest antimicrobial activity against the tested pathogenic and opportunistic strains was shown by LUHS244 and Bcur by-products. The CC formulation containing Gl and 5.0 g of Rasp by-products in combination with LUHS135 led to the best results in terms of TPC content, antioxidant activity, and LAB cell viability during 24 days of storage, as well as the highest OA and score for the EM ‘happy’ by consumers. However, the CCs with 5.0 g of Bcur by-products in combination with LAB showed slightly lower results for the tested parameters. Finally, Gl, Rasp and Bcur by-products, and MP-multiplied and lyophilised LAB are promising ingredients for the sustainable preparation of CCs. Further research is planned to assess the influence of antimicrobial ingredients on the normal microflora of the host. Taking into consideration that the digestive tract microbiota is related to the immune system and overall health status, these future steps are very important.

## Figures and Tables

**Figure 1 foods-11-01177-f001:**
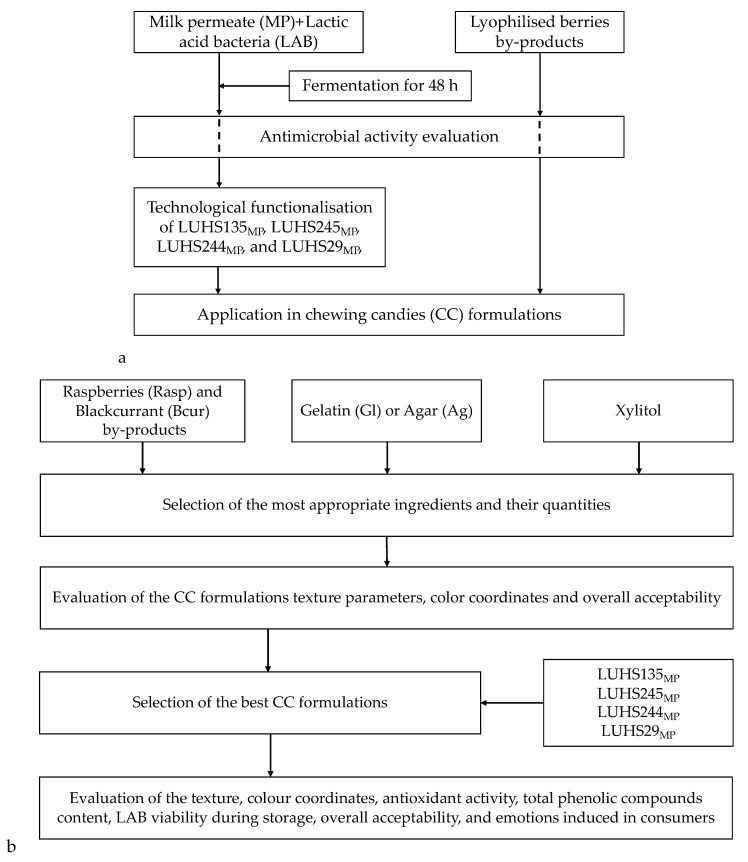
Principal scheme of experiment (**a**,**b**). CC—chewing candies; MP—milk permeate; LAB—lactic acid bacteria; Rasp—raspberry; Bcur—blackcurrant; Gl—gelatin; Ag—agar; LUHS135—*Lactiplantibacillus plantarum*; LUHS245—*Liquorilactobacillus uvarum*; LUHS244—*Lacticaseibacillus paracasei*; LUHS29—*Pediococcus acidilactici*.

**Figure 2 foods-11-01177-f002:**
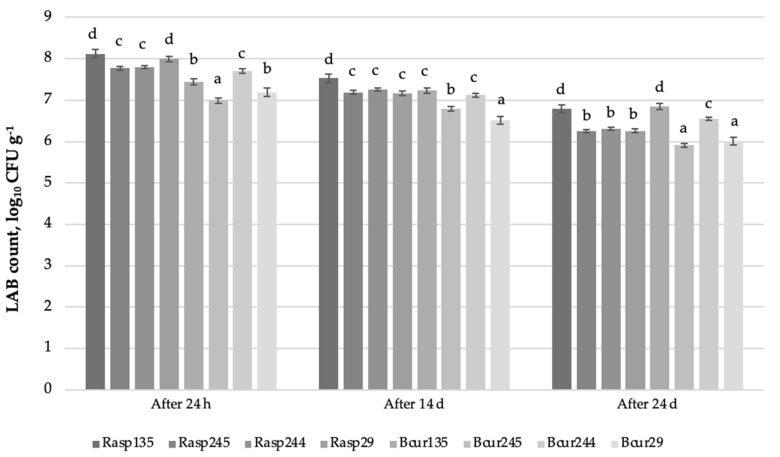
Viable lactic acid bacteria count in prepared nutraceutical candies during storage. Rasp—raspberry by-products; Bcur—blackcurrant by-products; 135—*Lactiplantibacillus plantarum* LUHS135; 245—*Liquorilactobacillus uvarum* LUHS245; 244—*Lacticaseibacillus paracasei* LUHS244; 29—*Pediococcus acidilactici* LUHS29; LAB—lactic acid bacteria; CFU—colony-forming units; ^a–d^ mean values with different superscript letters between columns are significantly different (*p* ≤ 0.05).

**Table 1 foods-11-01177-t001:** CC formulations with different gel-forming agents and berry by-products (the first stage of the experiment).

CC Formula	Gel-Forming Ingredient		Water, mL	Citric Acid, g	Xylitol, g	Berry By-Product	
	Gl, g	Ag, g				Rasp, g	Bcur, g
CC2.5raspGl	10.0	-	100.0	1.0	4.0	2.5	-
CC2.5raspAg	-	10.0				2.5	-
CC5.0raspGl	10.0	-				5.0	-
CC5.0raspAg	-	10.0				5.0	-
CC2.5bcurGl	10.0	-				-	2.5
CC2.5bcurAg	-	10.0				-	2.5
CC5.0bcurGl	10.0	-				-	5.0
CC5.0bcurAg	-	10.0				-	5.0

CC—chewing candies; Rasp—raspberry; Bcur—blackcurrant; Gl—gelatin; Ag—agar.

**Table 2 foods-11-01177-t002:** CC formulations based on the selected optimal quantities of berry industry by-products, selected LAB strains, and gelatin (the second stage of the experiment).

CC	Gl, g	Water, mL	Citric Acid, g	Xylitol, g	Berry By-Product, g		Lactic Acid Bacteria, g			
					Rasp	Bcur	LUHS135	LUHS245	LUHS244	LUHS29
Rasp135	10.0	100.0	1.0	4.0	5.0	-	5.0	-	-	-
Rasp245						-	-	5.0	-	-
Rasp244						-	-	-	5.0	-
Rasp29						-	-	-	-	5.0
Bcur135					-	5.0	5.0	-	-	-
Bcur245					-		-	5.0	-	-
Bcur244					-		-	-	5.0	-
Bcur29					-		-	-	-	5.0

CC—chewing candies; Gl—gelatin; Rasp—raspberry; Bcur—blackcurrant; LUHS135—*Lactiplantibacillus plantarum*; LUHS245—*Liquorilactobacillus uvarum*; LUHS244—*Lacticaseibacillus paracasei*; LUHS29—*Pediococcus acidilactici*.

**Table 3 foods-11-01177-t003:** Antimicrobial activity of LAB multiplied in MP and berry by-products.

Sample	Diameter of Inhibition Zone, mm									
	Pathogenic and Opportunistic Strains									
	Pat1	Pat2	Pat3	Pat4	Pat5	Pat6	Pat7	Pat8	Pat9	Pat10
	LAB multiplied in MP									
MP_LUHS135_	nd	nd	nd	nd	nd	nd	13.5 ± 0.6 ^a,b^	nd	10.2 ± 0.6	20.4 ± 0.6 ^b^
MP_LUHS245_	nd	nd	nd	nd	nd	nd	15.4 ± 0.3 ^b^	13.0 ± 0.6 ^b^	nd	20.2 ± 0.9 ^b^
MP_LUHS244_	nd	nd	nd	nd	nd	nd	15.3 ± 0.2 ^b^	10.2 ± 0.1 ^a^	nd	20.7 ± 0.6 ^b^
MP_LUHS29_	nd	nd	nd	nd	nd	nd	12.7 ± 0.4 ^a^	nd	nd	15.0 ± 0.1 ^a^
Berry by-products										
Rasp	nd	13.1 ± 0.3 ^a^	15.1 ± 0.3 ^b^	11.3 ± 0.4 ^a^	12.4 ± 0.2 ^a^	18.1 ± 0.4 ^a^	25.0 ± 0.5 ^b^	16.3 ± 0.4 ^a^	15.5 ± 0.3 ^a^	30.6 ± 0.4 ^a^
Bcur	10.2 ± 0.2	16.2 ± 0.1 ^b^	14.0 ± 0.2 ^a^	12.5 ± 0.1 ^b^	14.3 ± 0.3 ^b^	21.4 ± 0.2 ^b^	20.1 ± 0.1 ^a^	17.0 ± 0.3 ^a^	18.1 ± 0.4 ^b^	30.7 ± 0.5 ^a^

^a,b^ Mean values with different superscript letters between rows are significantly different (*p* ≤ 0.05); data expressed as the mean value (*n* = 3) ± standard deviation (SD). nd—not detected; LAB—lactic acid bacteria; MP_LUHS135_—*Lactiplantibacillus plantarum* LUHS135 multiplied in milk permeate; MP_LUHS245_—*Liquorilactobacillus uvarum* LUHS245 multiplied in milk permeate; MP_LUHS244_—*Lacticaseibacillus paracasei* LUHS244 multiplied in milk permeate; MP_LUHS29_—*Pediococcus acidilactici* LUHS29 multiplied in milk permeate; Rasp—lyophilised raspberry by-products; Bcur—lyophilised blackcurrant by-products. Pat1—*Salmonella enterica*; Pat2—*Pseudomonas aeruginosa*; Pat3—methicillin-resistant *Staphylococcus aureus* (MRSA) M87fox; Pat4—*Enterococcus faecalis*; Pat5—*E. faecium*; Pat6—*Bacillus cereus*; Pat7—*Streptococcus mutans*; Pat8—*Staphylococcus epidermis*; Pat9—*Staphylococcus haemolyticus*; Pat10—*Pasteurella multocida*.

**Table 4 foods-11-01177-t004:** Texture, colour coordinates, and overall acceptability of chewing candies based on different quantities of berry by-products and different gel-forming agents.

Sample	Texture (Hardness), mJ	Colour Coordinates, NBS			Overall Acceptability
		L*	a*	b*	
CC2.5raspGl	1.30 ± 0.01 ^b^	38.0 ± 0.16 ^f^	25.8 ± 0.57 ^g^	14.6 ± 0.39 ^g^	8.31 ± 0.2 ^c^
CC2.5raspAg	9.20 ± 0.05 ^e^	42.4 ± 0.14 ^h^	19.8 ± 0.17 ^e^	10.4 ± 0.19 ^f^	5.42 ± 0.5 ^b^
CC5.0raspGl	0.700 ± 0.01 ^a^	34.2 ± 0.46 ^e^	22.1 ± 0.44 ^f^	10.4 ± 0.19 ^f^	9.43 ± 0.3 ^d^
CC5.0raspAg	7.40 ± 0.03 ^c^	39.9 ± 0.31 ^g^	20.0 ± 0.23 ^e^	9.55 ± 0.09 ^e^	4.11 ± 0.6 ^a^
CC2.5bcurGl	1.30 ± 0.01 ^b^	27.9 ± 0.26 ^b^	1.87 ± 0.08 ^b^	1.24 ± 0.03 ^b^	8.23 ± 0.3 ^c^
CC2.5bcurAg	8.30 ± 0.05 ^d^	30.5 ± 0.26 ^d^	7.95 ± 0.12 ^d^	1.46 ± 0.05 ^c^	6.35 ± 0.6 ^b^
CC5.0bcurGl	1.30 ± 0.02 ^b^	26.9 ± 0.44 ^a^	1.36 ± 0.02 ^a^	1.18 ± 0.02 ^a^	9.20 ± 0.4 ^d^
CC5.0bcurAg	10.1 ± 0.05 ^f^	29.3 ± 0.53 ^c^	5.58 ± 0.06 ^c^	1.92 ± 0.03 ^d^	6.11 ± 0.4 ^b^

^a–h^ Mean values with different superscript letters between rows are significantly different (*p* ≤ 0.05); data expressed as the mean value (*n* = 3) ± standard deviation (SD); CC—chewing candy; 2.5, 5.0—content of berry by-products (g); Rasp—raspberry by-products; Bcur—blackcurrant by-products; Ag—agar, Gl—gelatin. L*—lightness; a*—redness; b*—yellowness.

**Table 5 foods-11-01177-t005:** Texture parameters and colour coordinates of CCs containing lactic acid bacteria.

Sample	Texture, mJ	Colour Coordinates, NBS		
		L*	a*	b*
Rasp135	1.10 ± 0.02 ^a,b^	35.6 ± 0.10 ^f^	21.7 ± 1.33 ^f^	10.6 ± 0.17 ^e^
Rasp245	1.60 ± 0.01 ^c^	37.6 ± 0.15 ^g^	17.8 ± 0.18 ^d^	10.0 ± 0.10 ^c^
Rasp244	1.60 ± 0.01 ^c^	35.7 ± 0.21 ^g^	19.1 ± 0.32 ^e^	10.7 ± 0.21 ^f^
Rasp29	1.91 ± 0.02 ^d^	34.1 ± 0.09 ^e^	19.0 ± 0.13 ^e^	10.3 ± 0.10 ^d^
Bcur135	0.80 ± 0.01 ^a^	27.5 ± 0.12 ^c^	2.24 ± 0.12 ^a^	1.35 ± 0.03 ^a^
Bcur245	1.65 ± 0.02 ^c^	27.6 ± 0.03 ^d^	2.91 ± 0.09 ^c^	1.54 ± 0.03 ^b^
Bcur244	1.23 ± 0.03 ^b^	27.0 ± 0.06 ^a^	2.07 ± 0.14 ^a^	1.36 ± 0.03 ^a^
Bcur29	2.10 ± 0.01 ^d^	27.3 ± 0.09 ^b^	2.45 ± 0.03 ^b^	1.34 ± 0.02 ^a^
Images of the CCs				
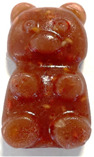	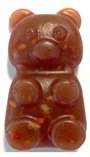	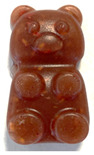	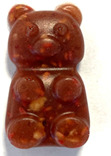	
Rasp135	Rasp245	Rasp244	Rasp29	
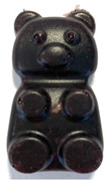	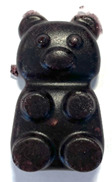	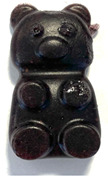	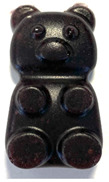	
Bcur135	Bcur245	Bcur244	Bcur29	

^a–g^ Mean values with different superscript letters between rows are significantly different (*p* ≤ 0.05); data expressed as the mean value (*n* = 3) ± standard deviation (SD). Rasp—raspberry by-products; Bcur—blackcurrant by-products; 135—*Lactiplantibacillus plantarum* LUHS135; 245—*Liquorilactobacillus uvarum* LUHS245; 244—*Lacticaseibacillus paracasei* LUHS244; 29—*Pediococcus acidilactici* LUHS29; L*—lightness; a*—redness; b*—yellowness.

**Table 6 foods-11-01177-t006:** Total phenolic compound (TPC) content and antioxidant activity of CCs containing lactic acid bacteria.

Sample	TPC, mg 100 g^−1^ d.m.	Antioxidant Activity, %
Rasp135	147.1 ± 0.13 ^h^	88.2 ± 1.34 ^e^
Rasp245	131.4 ± 1.02 ^f^	86.1 ± 0.95 ^e^
Rasp244	115.0 ± 0.75 ^d^	82.9 ± 1.12 ^d^
Rasp29	104.2 ± 0.37 ^c^	80.0 ± 1.05 ^c,d^
Bcur135	140.6 ± 0.56 ^g^	79.1 ± 1.45 ^c^
Bcur245	84.1 ± 0.15 ^a^	75.8 ± 0.86 ^b^
Bcur244	102.1 ± 0.43 ^b^	65.3 ± 0.77 ^a^
Bcur29	127.5 ± 0.96 ^e^	87.0 ± 1.02 ^e^

^a–h^ Mean values with different superscript letters between rows are significantly different (*p* ≤ 0.05); data expressed as the mean value (*n* = 3) ± standard deviation (SD). Rasp—raspberry by-products; Bcur—blackcurrant by-products; 135—*Lactiplantibacillus plantarum* LUHS135; 245—*Liquorilactobacillus uvarum* LUHS245; 244—*Lacticaseibacillus paracasei* LUHS244; 29—*Pediococcus acidilactici* LUHS29; TPC—total phenolic compound content.

**Table 7 foods-11-01177-t007:** Overall acceptability (OA) and emotions (EMs) induced in consumers by the prepared CCs.

Sample	Emotion (from 0 to 1)									
	OA	Neutral	Happy	Sad	Angry	Surprised	Scared	Disgusted	Contempt	Valence
Rasp135	9.52 ± 0.10 ^d^	0.390 ± 0.020 ^b^	0.231 ± 0.003 ^g^	0.081 ± 0.010 ^a^	0.055 ± 0.001 ^a^	0.011 ± 0.001 ^a^	0.002 ± 0.0002 ^b^	0.001 ± 0.0002 ^a^	0.100 ± 0.010 ^g^	0.421 ± 0.020 ^d^
Rasp245	7.51 ± 0.20 ^c^	0.531 ± 0.040 ^d^	0.162 ± 0.03 ^f^	0.141 ± 0.011 ^b^	0.233 ± 0.003 ^f^	0.021 ± 0.002 ^b^	0.010 ± 0.002 ^d^	0.003 ± 0.0002 ^c^	0.021 ± 0.006 ^c^	0.242 ± 0.030 ^b^
Rasp244	7.92 ± 0.30 ^c^	0.572 ± 0.061 ^d^	0.162 ± 0.002 ^f^	0.236 ± 0.006 ^e^	0.144 ± 0.001 ^e^	0.070 ± 0.004 ^e^	0.006 ± 0.0001 ^c^	0.006 ± 0.0002 ^d^	0.062 ± 0.004 ^f^	0.315 ± 0.020 ^c^
Rasp29	8.44 ± 0.20 ^c^	0.622 ± 0.031 ^e^	0.131 ± 0.001 ^e^	0.142 ± 0.010 ^b^	0.050 ± 0.004 ^a^	0.120 ± 0.002 ^f^	0.001 ± 0.0001 ^a^	0.001 ± 0.0001 ^a^	0.011 ± 0.002 ^b^	0.316 ± 0.001 ^c^
Bcur135	9.39 ± 0.20 ^d^	0.341 ± 0.033 ^b^	0.102 ± 0.001 ^d^	0.162 ± 0.012 ^b,c^	0.062 ± 0.001 ^b^	0.063 ± 0.001 ^d^	0.001 ± 0.001 ^a^	0.001 ± 0.0002 ^a^	0.040 ± 0.003 ^d^	0.260 ± 0.040 ^b^
Bcur245	5.30 ± 0.51 ^a^	0.120 ± 0.002 ^a^	0.023 ± 0.001 ^a^	0.221 ± 0.002 ^d^	0.170 ± 0.002 ^e^	0.010 ± 0.001 ^a^	0.001 ± 0.0002 ^a^	0.006 ± 0.0001 ^d^	0.055 ± 0.002 ^e^	0.150 ± 0.020 ^a^
Bcur244	6.57 ± 0.4 ^b^	0.436 ± 0.002 ^c^	0.046 ± 0.001 ^b^	0.185 ± 0.021 ^c^	0.111 ± 0.010 ^d^	0.010 ± 0.001a	0.001 ± 0001 ^a^	0.002 ± 0.0001 ^b^	0.010 ± 0.001 ^b^	0.191 ± 0.020 ^a, b^
Bcur29	7.31 ± 0.40 ^b,c^	0.651 ± 0.040 ^e^	0.070 ± 0.002 ^c^	0.090 ± 0.030 ^a^	0.081 ± 0.001 ^c^	0.040 ± 0.001 ^c^	0.001 ± 0001 ^a^	0.006 ± 0.0001 ^d^	0.001 ± 0.0002 ^a^	0.321 ± 0.030 ^c^

^a–g^ Mean values with different superscript letters between rows are significantly different (*p* ≤ 0.05). OA—overall acceptability (score); Rasp—raspberry by-products; Bcur—blackcurrant by-products; 135—*Lactiplantibacillus plantarum* LUHS135; 245—*Liquorilactobacillus uvarum* LUHS245; 244—*Lacticaseibacillus paracasei* LUHS244; 29—*Pediococcus acidilactici* LUHS29.

## Data Availability

The data are available from the corresponding author upon reasonable request.

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
