# Peer review of "Berry By-Products in Combination with Antimicrobial Lactic Acid Bacteria Strains for the Sustainable Formulation of Chewing Candies"

_foods, 2022, doi:10.3390/foods11091177_

Round 1

Reviewer 1 Report

Dear Authors,

your researches describe an interesting issue concerning novel food and new food formulations. Biologically active substance are of special interest to consumers. Researches are well described and the research plan is consistent and clearly presented. The major disadvantage of the manuscript concept is the postive effect of LAB present in chewing candies on human health. You postulate this effect, but used in the studies LAB strains are not probiotic. How do you explain this? On the other hand high carbohydrate content in CC can inhibit some pathogens anyway in the product. Do you agree?

I have some additional suggestions that I think can improve manuscript:

  • You use many abbreviations in the text, I suggest to list them in Abbreviation section;
  • Please write species names in italics (e.g. line 168);
  • Line 183 - "substance" - please specify which substance do you mean?
  • Lines 191, 193 - please describe more details on methods from references 36 and 37;
  • Table 6 - I have some doubts about the post-hoc test used to compare results of TPC, maybe you should use some more conservative test like Scheffe'a;
  • Don't you think that control CC should have been made without Bcur snd Rasp and second type of CC without LABs?

Author Response

Reviewer 1: 

your researches describe an interesting issue concerning novel food and new food formulations. Biologically active substance are of special interest to consumers. Researches are well described and the research plan is consistent and clearly presented. The major disadvantage of the manuscript concept is the postive effect of LAB present in chewing candies on human health. You postulate this effect, but used in the studies LAB strains are not probiotic. How do you explain this? On the other hand high carbohydrate content in CC can inhibit some pathogens anyway in the product. Do you agree?

Authors response: Authors are thankful for valuable comments. Yes, we agree with Reviewer 1, in this study used LAB strains are not probiotics. However, they possesses desirable antimicrobial and technological properties. Also, our previous studies showed, that L. uvarum can be used as an immune system modulators (https://doi.org/10.3390/foods10061313).

Regarding the carbohydrate content, we would like to explain, that fructose and glucose, show antimicrobial activity in high concentrations, however, berries by-products antimicrobial activity better can be explained by act of polyphenols and (or) organic acids.  Also, in the experiment we used berry by-products after juice preparation, so carbohydrate content in these substances, were low.

Moreover, information about antimicrobial activity of MP is scarce, but our previous studies showed, that during fermentation LAB, lactose is converted to GOS and organic acids, which also possesses antimicrobial activity. Finally, we would like to explain, that multifunctional composition of antimicrobial properties possessing ingredients was suggested, however, we agree with Reviewer 1, that further research is needed to identify, which compounds are the most important.

I have some additional suggestions that I think can improve manuscript:

Reviewer 1: You use many abbreviations in the text, I suggest to list them in Abbreviation section;

Authors response: Corrected.

 Abreventions:

Ag – agar

ANOVA - multivariate analysis of variance

Bcur - blackcurrant

CC - chewing candies

CFU – colony forming units

DIZ – diameter of inhibition zone

d.m. – dry matter

EMs – emotions

Gl – gelatin

GOS - galacto-oligosaccharides

LAB – lactic acid bacteria

LUHS135 – Lactiplantibacillus plantarum

LUHS244 – Lacticaseibacillus paracasei

LUHS245 – Liquorilactobacillus uvarum

LUHS29 – Pediococcus acidilactici

MP – milk permeate

MRS - The Man Rogosa and Sharpe

Na2CO3 – sodium carbonate

nd – not detected

OA – overall acceptability

Rasp - raspberry

SD - standard deviation

TPC - total phenolic compound

Reviewer 1: Please write species names in italics (e.g. line 168);

Authors response: Corrected.

Reviewer 1: Line 183 - "substance" - please specify which substance do you mean?

Authors response: Corrected. Wells of 6 mm in diameter were punched in the agar and filled with 50 μL of LAB multiplied in MP and 50 μg of Rasp and Bcur by-products

Reviewer 1: Lines 191, 193 - please describe more details on methods from references 36 and 37;

Authors response: Corrected. According improved citation, number of references, also changed from 36 to 43, and from 37 to 44.

0.2 mL of every fraction of free phenolics was blended with 1 mL of the Folin-Ciocalteau reagent and 0.8 mL of a saturated Na2CO3 solution. Prepared mixed solution was stored at room temperature (25°C) for 30 min and then the absorbance was measured at 765 nm in a spectrophotometer J.P. SELECTA S.A. V-1100D (Barcelona, Spain), and TPC content was expressed as microgram of gallic acid equivalent mL-1 of solution [43].

2 mL of DPPH solution (0.1 mM, in ethanol) was blended with 2 mL of the samples dissolved in the extracting solvent. The solution was shaken and incubated in the dark at room temperature (25 °C) for 60 min, and then the absorbance was measured at 517 nm in a spectrophotometer J.P. SELECTA S.A. V-1100D (Barcelona, Spain). The inhibition of the DPPH radical by the sample was calculated according to the following formula[44]:

Reviewer 1: Table 6 - I have some doubts about the post-hoc test used to compare results of TPC, maybe you should use some more conservative test like Scheffe'a;

Authors response: Authors are thankful for comment. We would like to explain, that usually Scheffe'a test is used with larger samples sizes. According to other publications (https://doi.org/10.1080/19476337.2018.1433721; https://doi.org/10.3390/foods10040777), authors used the post-hoc test to compare results of TPC. However, to compare TPC results, both tests could be used, and we would like to leave results statistic as it is presented.

Reviewer 1: Don't you think that control CC should have been made without Bcur snd Rasp and second type of CC without LABs?

Authors response:

Authors are thankful for comment.  We would like to explain, that the main idea was to develop multifunctional combinations, which includes optimal quantities of the Bcur and Rasp and LABs. We agree with Reviewer 1, however, these types of controls were not prepared. Agar and gelatin biased gels without other ingredients are not acceptable for consumers.  

Reviewer 2 Report

The work concerns the study of the production, in a sustainable way, of candies using lactic acid bacteria with antimicrobial activity in combination with berry by-products. In my opinion, the argument is very interesting. The experimentation was well conducted and the analyses carried out to determine the best formulation were appropriate. Results are clearly presented and the entire paper is well written. Considering this and the few comments listed below, I would strongly recommend the manuscript for minor revision.

  • The genera Lactobacillus nomenclature has been revised. The Authors should use the new one
  • Check italicum (i.e. LL 94, 292, 293, Tables 2 and 5)
  • The Authors should improve the conclusions, mainly explaining better the practical impact of the research results and indicating some possible useful investigations

Author Response

Reviewer 2: The work concerns the study of the production, in a sustainable way, of candies using lactic acid bacteria with antimicrobial activity in combination with berry by-products. In my opinion, the argument is very interesting. The experimentation was well conducted and the analyses carried out to determine the best formulation were appropriate. Results are clearly presented and the entire paper is well written. Considering this and the few comments listed below, I would strongly recommend the manuscript for minor revision.

Authors response: Authors are thankful for valuable comments.

Reviewer 2: The genera Lactobacillus nomenclature has been revised. The Authors should use the new one

Authors response: Corrected. Lactobacillus nomenclature has been changed: Lactiplantibacillus plantarum  Liquorilactobacillus uvarum; Lacticaseibacillus paracasei.

Reviewer 2: Check italicum (i.e. LL 94, 292, 293, Tables 2 and 5)

Authors response: Corrected.

Reviewer 2: The Authors should improve the conclusions, mainly explaining better the practical impact of the research results and indicating some possible useful investigation

Authors response: conclusions were corrected:

 Further research is planned to assess the influence of antimicrobial ingredients on normal microflora of the host. These future steps are very important, taking in to consideration, that digestive tract microbiota is related with the imune system and overall health status.

Reviewer 3 Report

The paper entitled ‘’Berry By-products in Combination with Antimicrobial Lactic Acid Bacteria Strains for Chewing Candies Formulation in Sustainable Manner’’ treats an interesting topic related to formulations of chewing candies by using raspberry and blackcurrant by-products in combination with lactic acid bacteria strains technological functionalised in milk permeate. 

There are some adjustments that need to be made:

Point 1 (Abstract):

1.1. Lines 32 - 33: ‘’The highest OA was shown for the CC prepared with gelatin and 5 g of Rasp and Bcur by-products.’’ Although the Abstract concisely describe the results and data interpretation, the amount of lyophilised berry by-products used should be expressed so that to be ease reproducible, being difficult to understand what exactly those 5 g of Rasp and Bcur by-products are reported to.

Point 2 (Material and Methods):

2.1. Paragraph 2.1. Materials Used for Preparation of CC

Please provide in brief the characteristics of the milk permeate, lyophilised Rasp and Bcur, agar powder and gelatin respectively, with focus on their parameters (physical and / or chemical) influencing the experimental.

2.2. Paragraph 2.3. Principal Scheme of the Experiment and Formulations of CC Recipes

In order to ensure the reproducibility of the experiment, please indicate the temperature of water in which the agar and gelatin respectively were soaked. Also, the level of the heating temperature of the soaked Ag, during those 5 min. before adding the other ingredients is required to be mentioned.

Point 3 (Results):

Paragraph 3.1. Evaluation of Antimicrobial Activity of LAB Multiplied in MP, and Berry By-Products

3.1. Could it be possible including some information on the level of LAB multiplication in MP after fermentation for 48 h at 30 °C? There are differences between the growth of LUHS135, LUHS244, LUHS245, and LUHS29 strains respectively?

3.2. Numerous differences were reported between the antimicrobial activity of lactic acid bacteria (Supplementary file 1) and the antimicrobial activity of lactic acid bacteria multiplied in MP (Table 1) against the same pathogenic and opportunistic strains. How are these ones explained?

3.3. Could it be possible including some information on the influence of the technological functionalisation of the LAB strains on their antimicrobial activity? Why did you choose to evaluate the antimicrobial activity of LAB multiplied in MP instead of the antimicrobial activity of the lyophilised LAB powder?

Paragraph 3.3. Evaluation of Texture and Colour Coordinates of CC Containing LAB

3.4. Please characterize in brief the lyophilised LAB biomass used for CC preparation (i.e. in terms of particle size) or explain why ‘’the addition of LUHS245, LUHS244, and LUHS29 increased the hardness of CC, while LUHS135 resulted in samples with a softer texture’’ (lines 360 - 361).

3.5. The images of CC are provided in Table 5. Could it be possible to explain what are the small particles visible inside the CC and if these ones influenced the OA of CC?

Paragraph 3.5. LAB Count in CC During Storage

3.6. Could it be possible thoroughly explain what ‘’nutraceutical candies’’ means so that to avoid any ambiguity?

Author Response

Reviewer 3: The paper entitled ‘’Berry By-products in Combination with Antimicrobial Lactic Acid Bacteria Strains for Chewing Candies Formulation in Sustainable Manner’’ treats an interesting topic related to formulations of chewing candies by using raspberry and blackcurrant by-products in combination with lactic acid bacteria strains technological functionalised in milk permeate. 

Authors response: Authors are thankful for valuable comments.

Reviewer 3: 

  • Lines 32 - 33: ‘’The highest OA was shown for the CC prepared with gelatin and 5 g of Rasp and Bcur by-products.’’ Although the Abstract concisely describe the results and data interpretation, the amount of lyophilised berry by-products used should be expressed so that to be ease reproducible, being difficult to understand what exactly those 5 g of Rasp and Bcur by-products are reported to.

Authors response: Corrected: 5 g of Rasp and 5 g of Bcur by-products

Point 2 (Material and Methods):

Reviewer 3.

2.1. Paragraph 2.1. Materials Used for Preparation of CC

Please provide in brief the characteristics of the milk permeate, lyophilised Rasp and Bcur, agar powder and gelatin respectively, with focus on their parameters (physical and / or chemical) influencing the experimental.

Authors response: Corrected. Ag powder form high gel strength at low concentrations, transparency and low viscosity in solution [39]. Gelatin forms a gel with cohesive internal structure, as well as have higher viscosity, melting and setting temperature [40].

Rasp and Bcur by-products, consisted of the seeds, peel and fibres were generated during wine production.

During fermentation, LAB convert lactose to prebiotic - galactooligosaccharides, however, information about MP physicochemical parameters is scarce [28].

  1. Zokaityte, E.; Cernauskas, D.; Klupsaite, D.; Lele, V.; Starkute, V.; Zavistanaviciute, P.; Ruzauskas, M.; Gruzauskas, R.; Juodeikiene, G.; Rocha, J.M.; et al. Bioconversion of Milk Permeate with Selected Lactic Acid Bacteria Strains and Apple By-Products into Beverages with Antimicrobial Properties and Enriched with Galactooligosaccharides. Microorganisms 2020, 8, 1182, doi:10.3390/microorganisms8081182.
  2. Xiao, Q.; Wang, X.; Zhang, J.; Zhang, Y.; Chen, J.; Chen, F.; Xiao, A. Pretreatment Techniques and Green Extraction Technologies for Agar from Gracilaria Lemaneiformis. Mar Drugs 2021, 19, 617, doi:10.3390/md19110617.
  3. Lee, K.Y.; Shim, J.; Bae, I.Y.; Cha, J.; Park, C.S.; Lee, H.G. Characterization of Gellan/Gelatin Mixed Solutions and Gels. LWT - Food Science and Technology 2003, 36, 795–802, doi:10.1016/S0023-6438(03)00116-6.

Reviewer 3. 2.2. Paragraph.  2.3. Principal Scheme of the Experiment and Formulations of CC Recipes

In order to ensure the reproducibility of the experiment, please indicate the temperature of water in which the agar and gelatin respectively were soaked. Also, the level of the heating temperature of the soaked Ag, during those 5 min. before adding the other ingredients is required to be mentioned.

Authors response:  For the preparation of CC with Ag, firstly, Ag powder was soaked in water for 30 min at 30± 2°C and melted by heating for 5 min till 100 °C temperature. Thereafter, other ingredients (Table 1) were added and mixed under boiling conditions (100  °C, while stirring). For the preparation of CC with Gl, firstly, Gl powder was soaked in water for 30 min at 30± 2°C temperature, and melted at 80 ± 2 °C.

Paragraph 3.1. Evaluation of Antimicrobial Activity of LAB Multiplied in MP, and Berry By-Products

Reviewer 3. 3.1. Could it be possible including some information on the level of LAB multiplication in MP after fermentation for 48 h at 30 °C? There are differences between the growth of LUHS135, LUHS244, LUHS245, and LUHS29 strains respectively?

Authors response: Corrected. The antimicrobial activity of LUHS135, LUHS245, LUHS244, and LUHS29 is shown in Supplementary File S1 and our previous studies showed, that LUHS135, LUHS244, LUHS245 and LUHS29 count in MP, 8.58 ± 0.24, 8.06 ± 0.31, 8.68 ± 0.39, and 8.19 ± 0.23 log10 CFU g-1, respectively, after multiplication for 48 h at 30 °C, were found [28].

Reviewer 3. 3.2. Numerous differences were reported between the antimicrobial activity of lactic acid bacteria (Supplementary file 1) and the antimicrobial activity of lactic acid bacteria multiplied in MP (Table 1) against the same pathogenic and opportunistic strains. How are these ones explained?

Authors response: Authors are thankful for comment. We would like to explain, that different LAB has a different metabolic profile in different substrates, it could be, that different metabolites were formed, which showed different inhibition properties. However, LAB metabolites were not analysed in this study.

Reviewer 3. 3.3. Could it be possible including some information on the influence of the technological functionalisation of the LAB strains on their antimicrobial activity? Why did you choose to evaluate the antimicrobial activity of LAB multiplied in MP instead of the antimicrobial activity of the lyophilised LAB powder?

Authors response: Authors are thankful for comment. We would like to explain, that before lyophilization, LAB must be somewhere multiplied. Unfortunately, commercial broths are very expensive and must be eliminated from the biomass before it uses in food industry. For this reason, MP for LAB biomass growth was used. In addition to more sustainable and cheaper biomass preparation, the tested LAB strains showed characteristics to convert lactose to GOS. Finally, double addition value was obtained: (I) commercial substrates was changed to the dairy industry by-products and (II) in addition to high number of LAB, GOS were produced.

Paragraph 3.3. Evaluation of Texture and Colour Coordinates of CC Containing LAB

Reviewer 3. 3.4. Please characterize in brief the lyophilised LAB biomass used for CC preparation (i.e. in terms of particle size) or explain why ‘’the addition of LUHS245, LUHS244, and LUHS29 increased the hardness of CC, while LUHS135 resulted in samples with a softer texture’’ (lines 360 - 361).

Authors response: Authors are thankful for comment, we would like to explain, that in this experiment, we do not use pure LAB, they were multiplied in MP and later lyophilised. We think, that substrate and specie of LAB could have influence on CC texture, however the data in the other scientific publications according L. plantarum influence on this parameter for CC is scarce.

Reviewer 3. 3.5. The images of CC are provided in Table 5. Could it be possible to explain what are the small particles visible inside the CC and if these ones influenced the OA of CC?

Authors response: Authors are thankful for valuable comment and we would like to explain, that these small particles visible inside the CC is seeds of raspberry by-products (samples Rasp135; Rasp244; Rasp245 and Rasp29) as well as particles of lyophilised lactic acid bacteria.  According to results of overall acceptability (OA), these ingredients have influence on OA, but not because of particles size or parts of raspberry by-products, but because of combined taste sensory.

Paragraph 3.5. LAB Count in CC During Storage

Reviewer 3. 3.6. Could it be possible thoroughly explain what ‘’nutraceutical candies’’ means so that to avoid any ambiguity?

Authors response: Corrected. According to Zokaityte et al. [7], Gl can be recommended for extending LAB viability in nutraceutical CC (term “nutraceutical” is used to describe medicinally or nutritionally functional foods), because, after 14 days of storage, they found LAB counts higher than 6.0 log10 CFU g−1 in samples prepared with Gl.

Round 2

Reviewer 1 Report

I would like to thank for all explanations. In my opinion your manuscript is acceptable in the present form.